# Proportions and determinants of successful surgical repair of obstetric fistula in low- and middle-income countries: A systematic review and meta-analysis

Liknaw Bewket Zeleke[1,2]*, Alec Welsh[2], Gedefaw Abeje[3], Marjan Khejahei[2,4,5,6]

1 College of Health Sciences, Debre Markos University, Debre Markos, Ethiopia, 2 Faculty of Medicine, School of Women's and Children's Health, University of New South Wales Sydney, Kensington, Australia, 3 School of Public Health, College of Medicine and Health Sciences, Bahir Dar University, Bahir Dar, Ethiopia, 4 Women's and Newborn Health, Westmead Hospital, Westmead, New South Wales, Australia, 5 Westmead Clinical School, University of Sydney, Sydney, Australia, 6 Western Sydney University, Sydney, Australia

* l.zeleke@unsw.edu.au

**Data Availability Statement:** The data supporting this study have been submitted as Supporting information (S5 excel sheet).

## Abstract

### Background

Obstetric fistula is a serious and debilitating problem resulting from tissue necrosis on the reproductive and urinary and/or lower gastrointestinal tract organs due to prolonged labor. Primary studies of the treatment of obstetric fistulae report significantly variable treatment outcomes following surgical repair. However, no systematic review and meta-analysis has yet estimated the pooled proportion and identified the determinants of successful obstetric fistula surgical repair.

### Objective

To estimate the proportion and identify the determinants of successful surgical repair of obstetric fistulae in low- and middle-income countries.

### Methods

The protocol was developed and registered at the International Prospective Register of Systematic Reviews (ID CRD42022323630). Searches of PubMed, Embase, CINAHL, Scopus databases, and gray literature sources were performed. All the accessed studies were selected with Covidence, and the quality of the studies was examined. Finally, the data were extracted using Excel and analyzed with R software.

### Results

This review included 79 studies out of 9337 following the screening process. The analysis reveals that 77.85% (95%CI: 75.14%; 80.56%) of surgical repairs in low and middle-income countries are successful. Women who attain primary education and above, are married, and have alive neonatal outcomes are more likely to have successful repair outcomes. In

**Funding:** The author(s) received no specific funding for this work.

**Competing interests:** The authors have declared that no competing interests exist.

contrast, women with female genital mutilation, primiparity, a large fistula size, a fistula classification of II and above, urethral damage, vaginal scarring, a circumferential defect, multiple fistulae, prior repair and postoperative complications are less likely to have successful repair outcomes.

## Conclusion

The proportion of successful surgical repairs of obstetric fistula in low and middle-income countries remains suboptimal. Hence, stakeholders and policymakers must design and implement policies promoting women's education. In addition, fistula care providers need to reach and manage obstetric fistula cases early before complications, like vaginal fibrosis, occur.

## Background

A fistula is an abnormal passageway that connects two organs or vessels that do not connect normally. Obstetric fistula refers to a fistula that occurs between reproductive tract organs (the vagina, cervix, and uterus), and lower urinary tract (bladder, urethra, and pelvic ureters) and/or lower gastrointestinal organs (rectum and anus) resulting from prolonged and/or obstructed labor or operative injury. This debilitating problem mainly occurs due to prolonged labor that causes tissue damage through tissue necrosis, leaving patients incontinent for both urine and feces. However, other risk factors such as lower gynecologic age, malnutrition, female genital mutilation, and living far from health facilities also contribute to the occurrence of obstetric fistula [1–5].

Obstetric fistula affects between 50,000 to 100,000 women worldwide each year and it is estimated that around two million women live with untreated obstetric fistula in Asia and Sub-Saharan Africa. It affects women at any age, but nearly half of the patients are youths aged between 10 and 19 years. The problem can be prevented by increasing access to high-quality maternal health care services, mainly access to safe perinatal care and childbirth services, improving the wealth status of the community, and by community education to avoid poor practices (early marriage, female genital mutilation, and home birth). Obstetric fistula been eliminated in the developed world by the improvement of access to maternal healthcare services [6–10].

Obstetric fistula is both a preventable and treatable public health problem. However, the majority of women with obstetric fistulae do not seek treatment due to multiple factors such as lack of awareness of treatment services, lack of access to transportation and the financial cost for the services, trying home remedies, and fear of stigma and social isolation. Treatment strategies for obstetric fistula may take the form of conservative treatment, surgical repair, physiotherapy, rehabilitation, or social reintegration techniques, depending on the severity of the fistula. Surgical repair aims to stop the incontinence, whereas the other treatment modalities aim to restore the patients' psychological well-being and reintegrate them into their families and society [8,11–17].

For surgical repair of the obstetric fistula, a successful treatment outcome refers to the successful closure of the fistula and cessation of the incontinence. Many individual studies have been published on the surgical repair outcomes of obstetric fistulae with a significant variation in the findings reported, ranging from a 42.5% [18] to 100% [19] closure rate, and a 39.66%

[20] to 90.1% [21] continence rate. According to the literature, the site, size, type and classification of the fistula, as well as the presence of scarring, surgeon's experience, previous attempts, type of repair technique, and duration of living with the fistula were the main determinants of the repair outcomes [22–24]. However, very few [25,26] systematic review and meta-analysis studies have been conducted on successful surgical repair outcomes and determinants of obstetric fistula but none of them covered low- and middle-income countries comprehensively. Investigating the pooled prevalence of successful treatment outcomes and determinant factors will help policymakers and program designers develop programs and guidelines to improve healthcare services for obstetric fistula patients. It will also give the scientific community an overview of obstetric fistula repair outcomes in low- and middle-income countries.

## Objective

This systematic review and meta-analysis study aims to estimate the pooled proportion of successful surgical repair outcomes of obstetric fistulae and determinants of the outcomes in low- and middle-income countries.

## Review questions

What is the pooled proportion of successful surgical obstetric fistula repair outcomes in low- and middle-income countries?

What are the determinants of successful surgical obstetric fistula repair outcomes in low- and middle-income countries?

## Methods

### Protocol design and registration

The title of the review was registered at the International Prospective Register of Systematic Reviews (PROSPERO) with registration ID CRD42022323630 on May 10, 2022. A protocol was developed, based on the Joanna Briggs Institute (JBI) evidence synthesis updated methodological guidelines for the conduct of systematic reviews and meta-analysis, and reported according to the Preferred Reporting Items for Systematic Review and Meta-analysis Protocols (PRISMA-P) [27] guidelines.

### Eligibility criteria

The inclusion and exclusion criteria were determined using the CoCoPop (condition, context, and population) framework, which JBI recommends for determining pooled prevalence and incidence for observational studies [28]. Accordingly, the *condition* refers to successful obstetric fistula surgical repair outcomes, *context* refers to low- and middle-income countries, and *population* refers to women who underwent surgical repair for obstetric fistula. The POE (population, outcome, and exposure) framework was considered in searching for studies on the determinants of obstetric fistula. The *exposure* in this context refers to the individual variables associated with the *outcome* variable, which is successful (yes/no) surgical repair of obstetric fistula.

All studies that reported treatment outcomes (either successful or unsuccessful) and/or determinants of obstetric fistulae, irrespective of publication status and year, were considered in the review. However, the review was limited to studies available in the English language and in low- and middle-income countries based on the World Health Organization (WHO) classification. More specifically, systematic reviews, case reports, case series, qualitative studies,

articles without full text, studies that involved non-obstetric fistula participants and studies from developed countries were excluded from this study.

## Search strategy and study selection

The search was conducted on databases and gray literature to access published and unpublished studies, including dissertations and preprints. The databases searched were PubMed, Embase, CINAHL, and Scopus, whereas Google Scholar and direct Google search were considered for the searching of gray literature. In addition, the first author contacted corresponding authors via publicly available ResearchGate accounts to access the full text of the abstracts that did not have full text, and one author [29] provided the full text of his article. Finally, the search was expanded by forward and backward citation searching to avoid missing relevant studies. The search was mainly conducted from June 19–20, 2023, and was open for continuous updates to include relevant new publications thereafter.

**Search terms.** Systematic searching was conducted after the identification of the domains of search terms according to the CoCoPop framework by taking successful outcomes of obstetric fistula surgical repair as conditions, low- and middle-income countries as context, and women who underwent surgical repair for obstetric fistula as the population. Initially, keywords were identified by breaking down the topic, then subject heading terms were explored, according to the nature of the databases. Finally, the keywords, subject heading terms, and free texts were combined to organize the search terms. Search terms within a similar domain were combined using the Boolean operator "OR" whereas the Boolean operator "and" was used to combine search terms from different domains. Truncation of words by an asterisk (*) was also applied to find words with various endings having the same root word. The details of the combinations of the search terms used in each database are attached *S 1 text*.

**Study selection.** All the accessed studies underwent a selection process according to the PRISMA flow chart (Appendix 2). The screening process was conducted using the Covidence screening tool. Initially, duplicated studies were removed, followed by screening of the title and abstract of the non-duplicated studies for their eligibility to be transferred to full-text review. Finally, relevant studies were identified via full-text review.

## Data extraction

The first author and the last author extracted the data to an Excel spreadsheet. The second author was invited to intervene in any disagreement between the two authors during data extraction. The data extraction spreadsheet contained the authors' names, publication year, study setting (country), study design, sampling technique, data collection method, data retrieval period, type of fistula, route of surgery, assessment period, event, and sample size of each included study. Then, relevant calculations such as the proportion of successful treatment outcomes, standard deviation, and standard error were generated through calculations made on the data extraction spreadsheet.

## Study quality and risk-of-bias assessment

The methodological quality of the selected studies was appraised at the study level using the JBI critical appraisal tools recommended for each study design [28,30]. Therefore, this review utilized the JBI critical appraisal tools designed for prevalence studies, analytical cross-sectional studies, cohort studies, case-control studies and randomized controlled trial studies. Two authors conducted the quality appraisal and scored the ratings independently as per the nature of the quality assessment tool. A third author also participated in quality appraisal upon request to resolve disagreements between the two authors.

**Risk of bias for individual studies.** The quality of studies assessed using JBI critical appraisal tools were rated as "Yes", "No", "Unclear" or "Not Applicable". Likewise, the quality of the randomized trial, which was evaluated using the Cochrane Collaboration tool, was rated as "high", "low" or "unclear" [31,32]. Finally, the rates were converted into percentages to categorize the risk of bias in each study. The quality of the included studies was classified as low quality (score ≤49%), moderate quality (score 50–69%), and high quality (Score ≥70) (106, 107).

**Risk of bias across studies.** Publication bias was assessed during the analysis graphically using a funnel plot and statistically using Egger and Begg tests. Furthermore, sensitivity analyses were conducted, which excluded outlier studies, studies with small sample sizes (n < 100), and studies that did not specify the fistula type and repair route.

## Data management and analysis

After completing the extraction, the data were transferred to R software for further management and analysis. The characteristics of the included studies were presented using descriptive statistics, tables, and figures. In addition, the statistical and figurative displays were supplemented with narrative descriptions. The pooled proportion of successful surgical repair outcomes of obstetric fistula was determined using the appropriate estimation model after the heterogeneity test and is presented using a forest plot.

The heterogeneity between the included studies was checked by a Cochran's Q test, quantified by the $I^2$ statistic value to choose either a random effect or fixed effect model. The presence of heterogeneity was confirmed when the probability value of the Cochran's Q test was less than 0.05. In addition, the quantified $I^2$ value (percentage) was interpreted as either low (below 30–40%), moderate (30–60%), considerably high (50–90%), or highly heterogenous (75–100%) [33]. If the $I^2$ indicated the presence of heterogeneity, the random effect analysis was chosen, whereas the fixed effect model was considered for low and no heterogeneity. The source of heterogeneity was further examined through subgroup analysis and meta-regression. Finally, factors with a *p*-value of 0.05 with 95% confidence interval were declared statistically significant determinants.

## Results

### Study selection

The initial search yielded 9337 studies (9249 from database search and 88 from gray literature). Then, 2334 duplicate studies were excluded, and 6540 studies were removed by title and abstract review, leaving 410 studies to be retrieved. Following the retrieval, 122 studies were assessed for eligibility, and finally, 79 studies were included in this review (Fig 1).

### Study characteristics

All the included studies were published articles with the publication year within the years 1971 to 2023 [15,18,20,21,24,29,34–106]. Of the included studies, 75 used data gathered from single countries, whereas four studies involved data from multiple countries, and more than one-quarter (eighteen) of the included studies were from Ethiopia. The review covered three WHO regions, namely Africa (56 studies), South-East Asia (eleven studies), Eastern Mediterranean (ten studies), and two studies involved data gathered from both South-East Asia and Africa. In terms of study design, nearly half (39) of the included studies were retrospective cross-sectional studies. Furthermore, most (72) of the included studies used surveys instead of sampling. A substantial number of studies did not specify some of the study characteristics such as

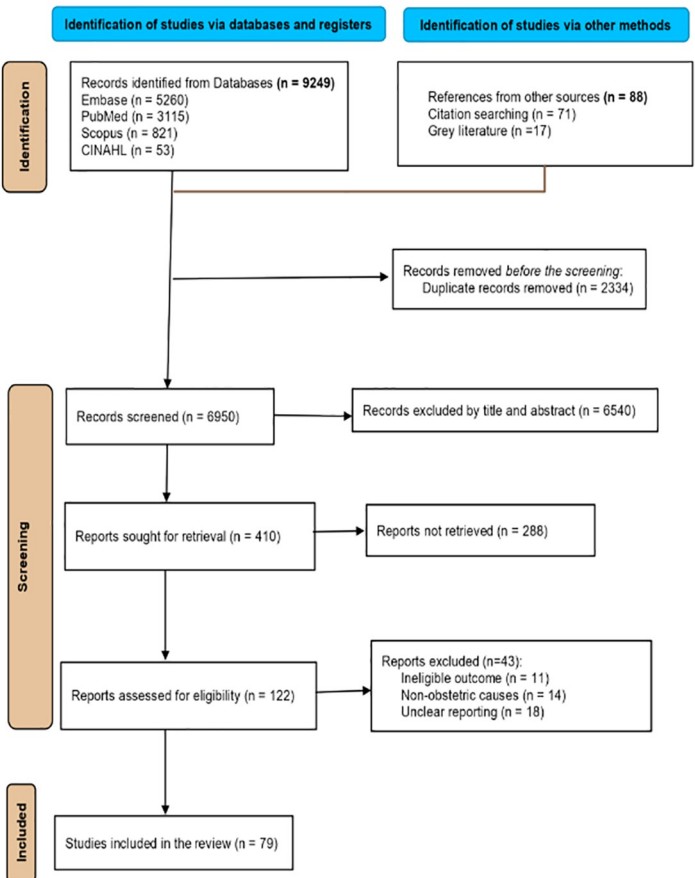

**Fig 1. PRISMA flow diagram.** This PRISMA flow diagram indicates the study screening process of the systematic review generated by Covidence software.

the data retrieval period (six studies), surgery route (26 studies), fistula type (eight studies), and outcome assessment period (nine studies). Finally, the pooled proportion of successful surgical repair of obstetric fistulae was estimated using data extracted from 77 studies; whereas the determinants were identified using data extracted from 28 studies. The detailed characteristics of the included studies have been presented in S1 Table.

## Risk of bias in studies

The risk of bias in each included study was assessed using the JBI critical appraisal tools appropriate for the design of the studies. The scores were converted into percentages to classify the risk of bias in each study into low, moderate, and high. The level of risk of bias for each study was valued inversely to the assigned study quality, i.e., studies with low quality were considered as having a high risk of bias, and studies with high quality were regarded as having a low risk of bias. Accordingly, 57 and 22 studies had a low and moderate risk of bias, respectively, and none of the included studies were rated to have a high risk of bias as presented in S2 Table.

**Proportion of successful surgical repair outcomes of obstetric fistula.** Following the selection and quality appraisal phases, 79 studies were transferred to the systematic review and meta-analysis phases to determine the pooled proportion of successful surgical repairs of obstetric fistulae and to identify the determinants of successful surgical repair. The minimum

proportion of successful recovery, 36%, was reported by Heller et al. [59] in Niger, and a 100% recovery was reported by Sharma et al. [46] in India. The pooled proportion of successful surgical repair of obstetric fistula was determined by meta-analysis among 76 studies, excluding the study by Sharma et al. [46] to avoid the undue outlier effect of 100% recovery. Therefore, the meta-analysis was conducted among a total of 25,664 observations, with a sample size per study ranging from 23 [90] to 2,360 [101]. The model of analysis was chosen after examining the heterogeneity test values (Fig 2).

*Heterogeneity test*. The presence of heterogeneity between the included studies was examined using Cochran's Q statistic and $I^2$ test values. Accordingly, the *p*-value of the Cochran's Q statistic ($< 0.01$) indicated the presence of significant between-study heterogeneity. Further quantification of the indicated heterogeneity using the $I^2$ test value (94.1%) implied the existence of considerably high heterogeneity among the included studies. Therefore, the random-effect model was chosen to determine the pooled proportion of successful repair of obstetric fistula. This model revealed a 77.85% [95%CI: 75.14; 80.56] rate of successful surgical repair. Given the presence of considerable heterogeneity, the possible source of the heterogeneity was examined through subgroup analysis and meta-regression.

*Subgroup analysis*. The subgroup analyses were performed by stratifying the pooled proportion by selected categorical variables such as WHO region, type of fistula, and repair route. The subgroup analysis by WHO region indicated a slightly lower within-group heterogeneity in Southeast Asia and the Eastern Mediterranean regions. However, the stratification by fistula type and surgical route showed the presence of high within-group heterogeneity in all the groups. The subgroup analysis by WHO region indicated the presence of significant between-group heterogeneity. In contrast, the subgroup analyses by fistula type and surgical route demonstrated a non-significant heterogeneity between the respective groups (Table 1).

*Meta-regression*. In addition to the subgroup analyses using the categorical characteristics of the studies, the possible source of heterogeneity by continuous characteristics, particularly the publication year and sample size, was examined through meta-regression analyses. However, the analyses showed that neither variable contributed to significant heterogeneity (Table 2).

*Sensitivity analysis*. Sensitivity analyses were conducted by excluding some studies considering the following criteria: outliers [18, 59, 94], studies with small sample size (n < 100), unspecified fistula type, and surgical route. However, none of the analyses significantly changed either the heterogeneity test or the pooled proportion of successful surgical repair (Table 3).

*Publication bias*. Possible publication bias was examined using a funnel plot, and the plot displayed an asymmetric distribution of the studies, which evidenced the presence of publication bias (Fig 3).

**Determinants of successful surgical repair outcomes of obstetric fistula.** The determinants of successful surgical repair outcomes of obstetric fistula were identified using a conceptual framework designed by Lewis Wall [107]. The framework was constructed to outline the determinants of obstetric fistula formation. Hence, the framework has been adapted with amendments to make it suitable for the aims of this review (Fig 4). A total of 27 studies were considered in the determinant identification analysis and the possible determinants of successful surgical repair were categorized into remote determinants of obstetric fistula, intermediate determinants of obstetric fistula, immediate determinants of obstetric fistula, fistula-related, and repair technique-related factors (Fig 4).

*Remote determinants of obstetric fistula*. This category mainly contains socio-demographic and cultural factors such as education, residence, occupation and income.

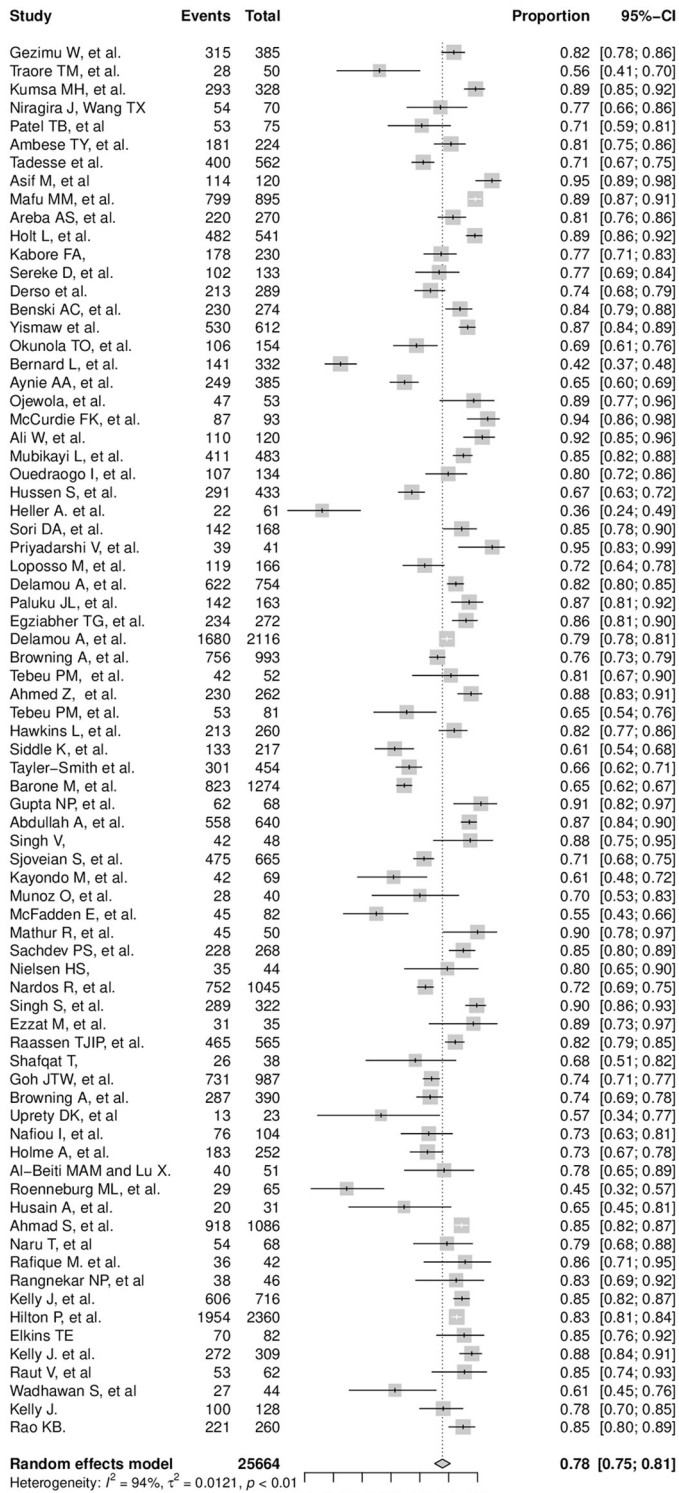

**Fig 2. Forest plot.** This forest plot indicates the pooled proportion of obstetric fistula surgical repair outcomes.

**Table 1. Subgroup analysis results by WHO region, fistula type, and surgical route.**

| Variable | Category | No. of studies | Proportion [95%CI] | Within-group I² | Between-group heterogeneity | |
|---|---|---|---|---|---|---|
| | | | | | Q | *p*-value |
| WHO region | Africa | 55 | 0.76 [0.73; 0.79] | 94.3% | 96.91 | < 0.01 |
| | South-East Asia | 10 | 0.83 [0.77; 0.90] | 73.7% | | |
| | Eastern Mediterranean | 10 | 0.86 [0.82; 0.91] | 75.1% | | |
| | South-East Asia and Africa | 1 | 0.65 [0.62; 0.67] | NA | | |
| Fistula type | VVF | 26 | 0.78 [0.72; 0.83] | 94.9% | 3.93 | 0.27 |
| | VVF and RVF | 25 | 0.77 [0.73; 0.82] | 94.3% | | |
| | Multiple | 17 | 0.82 [0.77; 0.86] | 92.9% | | |
| | Not specified | 8 | 0.73 [0.60; 0.86] | 94.4% | | |
| Surgical route | Vaginal | 11 | 0.77 [0.73; 0.82] | 84.6% | 9.51 | 0.049 |
| | Abdominal | 5 | 0.87 [0.75; 0.98] | 95.1% | | |
| | Vaginal and abdominal | 24 | 0.82 [0.78; 0.86] | 83.9% | | |
| | Vaginal, abdominal, and combined | 10 | 0.75 [0.63; 0.86] | 93.1% | | |
| | Not specified | 26 | 0.75 [0.69; 0.80] | 96.0% | | |

**Table 2. Meta-regression results by publication year and sample size.**

| Variable | I² | R² | Heterogeneity Test (Q [p-value]) | Model Results | |
|---|---|---|---|---|---|
| | | | | (Estimate [95% CI]) | (p-value) |
| Publication Year | 96.42% | 0.00% | 1276.54 [< 0.01] | -0.0004 [-0.0031, 0.0023] | 0.57 |
| Sample size | 96.32% | 0.00% | 1273.14 [< 0.01] | 0.0000 [-0.0001, 0.0001] | 0.63 |

**Table 3. Sensitivity analysis results by outliers, small sample size, unspecified fistula type and surgical route.**

| Exclusion criteria | I² | Heterogeneity Test (Q [p-value]) | Model | Pooled proportion [95% CI] |
|---|---|---|---|---|
| Outliers | 92.7% | 984.42 [< 0.01] | Random effect | 0.79 [0.77; 0.82] |
| Small sample size | 95.5% | 1041.18 [< 0.01] | Random effect | 0.79 [0.76; 0.82] |
| Unspecified fistula type | 94.2% | 1148.71 [< 0.01] | Random effect | 0.78 [0.76; 0.81] |
| Unspecified surgical route | 90.8% | 530.72 [< 0.01] | Random effect | 0.80 [0.77; 0.83] |
| **No studies excluded** | **94.4%** | **1170.78 [< 0.01]** | **Random effect** | **0.78 [0.75, 0.81]** |

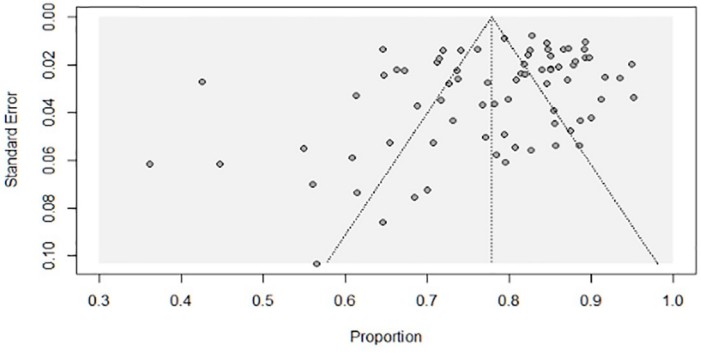

**Fig 3. A scatter plot.** This scatter diagram indicates the publication-bias distribution.

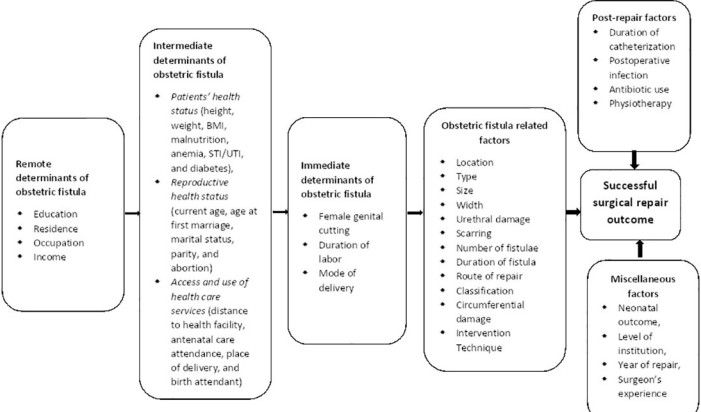

**Fig 4. A conceptual framework.** This conceptual framework shows the determinants of successful surgical repair outcomes of obstetric fistula [107].

*Intermediate determinants of obstetric fistula*. Includes the patients' health status (height, weight, BMI, malnutrition, anemia, sexually transmitted infections/urinary tract infection, and diabetes), reproductive status (current age, age at first marriage, marital status, parity, and abortion), access and use of health care services (distance to health facility, antenatal care attendance, place of delivery, and birth attendant).

*Immediate determinants of obstetric fistula*. The duration of labor and mode of delivery are the immediate factors according to the studies included in the systematic review.

*Fistula-related factors*. The location, type, size, width, urethral damage, scarring, number of fistulae, duration of fistula, route of repair, fistula classification, circumferential damage, and intervention technique are the fistula-related factors that might determine the successful surgical repair outcome.

*Post-repair factors*. These occur following the surgical repair of obstetric fistulae and include the duration and type of catheterization, postoperative infection, antibiotic use and physiotherapy.

*Miscellaneous factors*. This review identified some factors, namely neonatal outcome, level of institution, year of repair, and surgeon's experience, that are reported by the included studies as possible determinants of obstetric fistula surgical outcome. Given that these factors do not fit under the categories mentioned above, this review presents them as miscellaneous factors. Eligible factors that appeared in studies with consistent units of measurement or category were transferred to the meta-analysis. Therefore, the meta-analysis was conducted on age, height, marital status, education, residence, presence of comorbidity, female genital mutilation, parity, place of delivery, mode of delivery, labor duration, neonatal outcome, fistula type, fistula size, fistula classification, urethral damage, vaginal scarring, circumferential defect, multiple fistulae, route of repair, prior repair, and postoperative complications.

The meta-analysis indicated that successful surgical repair outcome of the obstetric fistula was significantly associated with marital status, female genital mutilation, parity, labor duration, neonatal outcome, fistula size, fistula classification, urethral damage, vaginal scarring, circumferential defect, multiple fistula, prior repair, and postoperative complications. An appropriate model of analysis was chosen for each factor after examining the heterogeneity test (Table 4).

*Remote determinants of obstetric fistula*. Based on the findings, the level of education showed positive association with the occurrence of successful fistula repair, while place of residence, occupation and income did not show a significant association at a 95% confidence

**Table 4. Determinants of successful surgical repair outcomes of obstetric fistula in low- and middle-income countries.**

| Variable (category) | OR [95% CI] | Model | Heterogeneity test | |
|---|---|---|---|---|
| | | | I²% | Q (p-value) |
| Age in years [40,48,51,77] | | | | |
| >20 | 1 | Random effects | | |
| <20 | 1.17 [0.47, 2.90] | | 62.3 | 5.30 (0.07) |
| Height in CM [34,36,48,49] | | | | |
| >150 | 1 | Random effect | | |
| ≤150 | 1.28 [0.66, 2.45] | | 64.0 | 8.33 (0.04) |
| Marital status [18,36,44,52,62,69] | | | | |
| Unmarried | 1 | Fixed effect | | |
| Married | 1.44 [1.14, 1.83]* | | 0.0 | 0.99 (0.96) |
| Education [18,34,36,48,52,62,69,71] | | | | |
| No formal education | 1 | Fixed effect | | |
| Primary and above | 1.80 [1.44, 2.26]* | | 21.2 | 7.61 (0.27) |
| Residence [36,39,48,52,62,71] | | | | |
| Urban | 1 | Random effect | | |
| Rural | 0.82 [0.52, 1.28] | | 64.9 | 13.47 (0.02) |
| Presence of comorbidity [44,49,66,71] | | | | |
| No | 1 | Fixed effect | | |
| Yes | 0.87 [0.63, 1.21] | | 0.0 | 2.20 (0.53) |
| Genital mutilation [39,49,71,76,77] | | | | |
| No | 1 | Random effect | | |
| Yes | 0.50 [0.32, 0.78]* | | 49.2 | 7.88 (0.10)[a] |
| Parity [21,36,39,40,49,52,64,74,77] | | | | |
| Multipara | 1 | Fixed effect | | |
| Primipara | 0.63 [0.53, 0.76]* | | 0.0 | 7.94 (0.44) |
| Delivery place[21,34,36,39,48,49,52] | | | | |
| Home | 1 | Random effect | | |
| Health facility | 1.61 [0.92,2.82] ᵀ | | 80 | 30.10 (<0.01) |
| Mode of delivery [36,39,44,48,49,52,62,69,71,78] | | | | |
| Spontaneous Vaginal | 1 | Random effect | | |
| Cesarean section and instrumental | 1.03 [0.79, 1.34] | | 43.6 | 15.97 (0.07) |
| Labor duration [21,36,39,40,44,48] | | | | |
| < 48 hours | 1 | Random effect | | |
| ≥ 48 hours | 0.57 [0.34, 0.95]* | | 82.3 | 28.19 (<0.01) |
| Neonatal outcome [36,39,40,44,49,62,69] | | | | |
| Stillbirth | 1 | Fixed effect | | |
| Alive | 2.3 [1.68, 3.13]* | | 34.9 | 9.22 (0.16)[b] |
| Fistula type [15,24,36,39,40,44,64] | | | | |
| Vesicovaginal | 1 | Random effect | | |
| Rectovaginal and mixed | 0.96 [0.63, 1.46] | | 50.5 | 12.13 (0.06) |
| Fistula size [21,34,36,37,39,40,44,53,56,71,76,88] | | | | |
| ≤3 CM | 1 | Random effect | | |
| >3 CM | 0.24 [0.17; 0.37]* | | 79.0 | 52.30 (<0.01) |
| Classification [21,40,44,51,52,69,74] | | | | |
| Type I | 1 | Fixed effect | | |
| Type II and above | 0.42 [0.33, 0.53]* | | 19.3 | 7.43 (0.28) |

(*Continued*)

**Table 4.** (Continued)

| Variable (category) | OR [95% CI] | Model | Heterogeneity test | |
|---|---|---|---|---|
| | | | I²% | Q (p-value) |
| Urethral status [34,40,48,62,71] | | | | |
| Intact | 1 | Fixed effect | | |
| Partially and totally damaged | 0.37 [0.31, 0.45]* | | 6.7 | 4.29 (0.37) |
| Vaginal scarring [21,37,39,40,51,56,62,71,76,77] | | | | |
| Mild | 1 | Random effect | | |
| Moderate to severe | 0.30 [0.16; 0.56]* | | 88.0 | 74.78 (0.00) |
| Circumferential defect [34,40,44,69,71,76,77] | | | | |
| No | 1 | Random effect | | |
| Yes | 0.30 [0.20, 0.44]* | | 60.2 | 15.06 (0.02) |
| Multiple fistula [40,53,71,74,78] | | | | |
| No | 1 | Random effect | | |
| Yes | 0.25 [0.13, 0.48]* | | 68.9 | 12.85 (<0.01) |
| Route of repair [36,37,44,52,53,64,73,74,78] | | | | |
| Vaginal | 1 | Random effect | | |
| Abdominal and mixed | 0.89 [0.68; 1.15] | | 38.9 | 13.08 (0.09) |
| Prior repair [24,39,52,53,56,64,69–71,74,76–78] | | | | |
| No | 1 | Random effect | | |
| Yes | 0.43 [0.35, 0.53]* | | 42.1 | 20.73 (0.05) |
| Postop complications [21,44,52,62] | | | | |
| No | 1 | Fixed effect | | |
| Yes | 0.40 [0.2, 0.60]* | | 0.0 | 0.62 (0.89) |

[a] Random effect model has been selected considering the **I²** despite the p-value of Q being > 0.05,

[b] Fixed effect has been selected considering the *p*-value of Q despite the **I²** showing minimal heterogeneity,

* Significantly associated with *p*-value < 0.05.

[T] *p*-value of fixed effect model < 0.05.

interval level. The odds ratio for the level of education was calculated using data obtained from eight studies. Accordingly, women who attained an educational level of primary and above were 80% (OR 1.80 [95% CI; 1.44, 2.26]) more likely to have successful surgical repair than women with no formal education.

*Intermediate determinants of obstetric fistula.* Married women were 44% (OR 1.44 [95% CI; 1.14, 1.83]) more likely to have successful surgical repair outcomes than women whose marital status was single, divorced, or widowed.

*Immediate determinants of obstetric fistula.* Women who had genital mutilation during childhood were 50% (OR 0.50 [95% CI; 0.32, 0.78]) less likely to attain successful surgical repair relative to women who had no genital cutting. Primipara women were 37% (OR 0.63 [95% CI; 0.53, 0.76]) less likely to have successful surgical repair than multipara women. Similarly, women with a longer duration of labor, particularly greater than two days and above, had 43% (OR 0.57 [95% CI; 0.34, 0.95]) less likely successful surgical repair outcomes than their counterparts.

*Fistula-related factors.* From fistula-related factors, women with larger fistula sizes (> 3cm) were 76% less likely to achieve successful surgical repair outcomes than women with small fistulae. Likewise, women whose fistulae were classified as type II and above based on Goh [21,38,42] and Waaldijk [49,50,67,74] classification methods were 58% less likely to have

successful surgical repair outcome than women with type I fistulae. Patients whose urethral status was partially or totally damaged were 63% less likely to achieve successful surgical repair than a patient who had an intact urethra. The study also revealed that patients with a moderate to severe forms of vaginal scarring were 73% less likely to have successful surgical repair outcomes than patients with mild scarring. In addition, patients with circumferential defects were also 70% less likely to have successful surgical repair outcomes than those without. Concerning the number of fistulae, women with multiple fistulae were 75% less likely to have successful repair outcomes than a woman with a single fistula. The study also demonstrated that patients with prior repair were 57% less likely to have successful surgical repair outcomes than patients undergoing the surgical repair for the first time.

*Postoperative and miscellaneous factors*. Patients with postoperative complications were 60% less likely to have successful surgical repair than patients without complications.

*Miscellaneous factors*. Patients who gave birth to a live neonate had a 2.3 times (OR 2.3 [95% CI; 1.68, 3.13]) more likely successful repair outcome than patients with stillbirth when the fistula occurred.

## Discussion

This systematic review and meta-analysis presents the review findings of 79 studies, of which 77 were used to determine the pooled proportion of successful surgical repair outcomes, and 28 were used to identify the determinants of successful obstetric fistula repair in low- and middle-income countries. The review included studies that measured the successful treatment of obstetric fistulae in terms of successful closure of the fistula and urinary and/or fecal continence achievement following the surgical repair.

The reviewed studies showed a substantially variable repair rate ranging from 36% [59] to 100% [46], which supported the necessity of determining the pooled proportion of obstetric fistula successful surgical repair through meta-analysis. Therefore, this study determined the pooled proportion and revealed a successful surgical repair outcome of 77.85% [95% CI: 75.14; 80.56] for obstetric fistulae in low and middle-income countries. This finding is higher than the finding of a systematic review and meta-analysis study [26] conducted by Kumsa et al. that determined the rate of successful obstetric fistula in Africa. This finding is also relatively comparable with another systematic review and meta-analysis by Ejigu et al. that gave a successful repair outcome of 73.11% in East Africa [25]. The current study's finding lies within the 95% confidence interval of Ejigu et al.'s study; however, our study confidence interval excludes the estimated proportion of the study mentioned above. This discrepancy might be due to the confidence interval variation of the two studies, secondary to the difference in the number of the included studies and in turn, the study participants. The study of Ejigu et al. was conducted among 16 studies comprising 6254 observations; the current meta-analysis estimated the pooled proportion using 25,664 observations from 76 studies. However, this study's pooled estimate is lower than the WHO standard: 85% fistula closure with 90% achieving continence [108], which implies the necessity of more coordinated and collaborative efforts toward eradicating obstetric fistula in low and middle-income countries.

This study also reviewed and identified the determinants of successful surgical repair of obstetric fistula. These determinants fall under the following themes: remote determinants of obstetric fistula, intermediate determinants of obstetric fistula, immediate determinants of obstetric fistula, fistula-related factors, and repair technique-related factors. The meta-analysis wing of this study identified that level of education, marital status, female genital mutilation, parity, neonatal outcome, fistula size, fistula classification, urethral damage, vaginal scarring, circumferential defects, multiple fistulae, prior repair and postoperative complications are

significantly associated (at 95% confidence interval level) with successful surgical repair outcomes of obstetric fistula.

Women who attained primary and above education had a better recovery outcome than those without formal education. This association might be attributed to early care-seeking behavior among women with better educational attainment [109], which prevents complications such as vaginal scarring that negatively affects successful repair outcomes.

Analysis of data from six studies [18,36,44,52,62,69] showed that married women were 44% more likely to have a successful surgical repair of obstetric fistula than their single counterparts. The positive impact of marital status on successful outcome might be justified by a husband's psychological and financial support, as indicated by studies that investigated the quality of life following the surgical repair of obstetric fistula patients [110–112]. However, none of the included studies demonstrate a significant association, clearly implying that a repeat study involving a large number of participants should be performed to investigate whether marital status determines the success of surgical repair of obstetric fistula.

The other important variable is female genital cutting. Collaborative efforts to end this practice are needed. Studies demonstrate an association between female genital cutting and obstetric fistula [107,113,114]. Even though this procedure is primarily performed during infancy and early childhood [115,116], it has a long-term health impact. This study proved that the health impacts of genital cutting persist even after surgical repair of obstetric fistula cases, resulting in poor recovery outcomes among patients with genital cutting compared with women with no genital cutting. The negative impact of this practice on the successful surgical repair of obstetric fistula might be due to the loss of elasticity in the genital parts due to scarring and fibrosis formation [117]. Women with genital cutting develop scars on the area where the excision has been performed, and this scar may hinder the wound healing process, leading to an unsuccessful repair outcome.

Regarding parity, primipara women had a lower likelihood of successful surgical repair than multipara women. This association might be attributed to the severity of the fistula [118]. Given that primipara women have no previous history of childbirth, the adequacy of the birth canal might be worse for the primipara women than for multiparas. A study in Ethiopia also reported that primipara women have longer labor durations, larger fistulae, more stillbirths, and more vaginal scarring than multipara women [119]. These conditions lead to experiencing a more complicated fistula and an unsuccessful surgical repair outcome in primipara women.

Similarly, women whose labor lasted 48 hours or more were less likely to have a successful surgical repair of the obstetric fistula, which might result from the greater complexity and severity or the injury. This finding is consistent with the result of a systematic review and meta-analysis study conducted in East Africa [25]. The fact that the main cause of obstetric fistula is necrosis of tissue due to occlusion of blood supply to the reproductive and/or lower gastrointestinal tract by the presenting part of the fetus, the more a mother stays in an obstructed labor, the more likely she is to develop a complicated and severe fistula.

This study also reported that women with livebirth neonatal outcomes were more likely to have successful surgical repair than those with stillbirth. The association might related to longer labor durations with obstructed and more complicated fistulae formation in women with stillbirth than those with livebirth, as evidenced by a systematic review and meta-analysis by Ayenew [120]. Furthermore, since childbirth is a highly anticipated life event for a woman and her family, experiencing a stillbirth imposes psychological distress on the woman [121], leading to unfavorable obstetric fistula surgical repair outcomes.

Regarding fistula size, women with large fistulae (> 3 cm) were less likely to have successful surgical repair as compared to women with fistula sizes less than 3 cm. A systematic review and meta-analysis study reported a similar finding in East Africa [25]. The possible

explanation might be that extensive damage leaves insufficient bladder tissue for a tension-free successful repair [122].

We also examined the association between fistula classification and successful repair outcome, despite the included studies lacking consistency in classification methods. Out of the seven studies included for the examination of the effect of classification on successful repair outcome, three studies used the Goh classification [21,40,44] method and four studies used the Waaldijk classification method [51,52,69,76]. We analyzed the findings of reports in both classifications after categorizing them into Grade II and above, and Grade I. The analysis revealed that women with Grade II and above classifications were less likely to have successful repair outcomes than women with Grade I fistulae. The possible explanation might be that both classification methods depend primarily on anatomic location, fistula size, scarring and vaginal length [123]. Hence, Grade II and above fistulae are more complicated and less likely to have successful surgical repair outcomes than Grade I [124].

In addition, obstetric fistulae accompanied by partial or total urethral damage are less likely to have successful surgical repair outcomes than those with intact urethras. This finding was supported by other studies [25,125]. The negative impact of urethral damage on successful surgical repair outcome might be due to the difficulty of repair in fistula cases complicated by urethral injury, particularly those requiring grafting [126]. Moreover, urethral damage in obstetric fistulae is among the main causes of persistent urinary incontinence due to injury to nerve endings [127]. Similarly, fistulae with circumferential defects were less likely to have a successful surgical repair than those without. This association might be justified similarly to the association with urethral damage because fistulae with circumferential defects are characterized by the destruction of the whole circumference of the bladder and detachment of the urethra from the bladder [128]. Overall, the closure rate of fistulae with urethral damage and circumferential defects may not differ from other types of fistulae, but maintaining functional continency remains the main challenge [129].

Women with moderate and severe forms of vaginal scarring are less likely to have successful repair outcome than women with mild or no scarring. This finding aligns with other systematic review and meta-analysis studies [25,125]. The possible reason might be a lack of vascularization and a poor healing process [130].

Furthermore, women with multiple fistulae and prior repair had a lower chance of successful surgical repair outcome than their counterparts. Other studies [109,126] support these results. In addition, women with postoperative complications are less likely to attain successful surgical repair outcomes than women without complications.

### Limitations of the reviewed studies

Some of the included studies have limitations that impacted the completeness of the data extraction. One of the main limitations was the use of studies that were missing relevant characteristics, mainly the data retrieval period, fistula type, route of repair, and outcome assessment period. These are indicated as unspecified in the study characteristics table. In addition, the included studies lack consistency in measurement and recoding. For instance, many studies have reported 'age' as a determinant variable. However, due to inconsistency in transforming this to a categorical variable, we excluded some of the studies in the meta-analysis in order to the association between age and successful surgical repair outcome.

### Strengths and limitations of this study

This study included a large number of studies and participants and used updated guidelines and manuals. Moreover, the study's title has been registered on PROSPERO, and a protocol

was developed prior to the main review. On the other hand, this study could not exclude non-obstetric cases included in some of the primary studies. In addition, this study excluded studies written in languages other than English and articles without full text, which would contain impactful data.

## Conclusions and recommendations

The rate of successful surgical repair of obstetric fistulae in low and middle-income countries remains low compared to the WHO recommendation. Level of education, marital status, female genital mutilation, parity, neonatal outcome, fistula size, fistula classification, urethral damage, vaginal scarring, circumferential defect, multiple fistulae, prior repair, and postoperative complications are the factors determining the successful surgical repair of obstetric fistulae. Hence, the authors recommend that stakeholders expand maternal healthcare service facilities and staffing in low-resourced settings to avoid long labor hours and provide timely advanced interventions, mainly terminating prolonged labor through cesarean section. The findings of this study also imply that implementing policies that promote women's education enhances the successful surgical repair of obstetric fistulae. Government and non-government stakeholders should strengthen their endeavors to eliminate female genital mutilation, since it remains a significant etiologic and determinant factor for successful treatment outcomes. Fistula care-service providers also need to design and implement strategies to reach out to patients before the health issues become complicated, such as patients developing fibrosis. Finally, further research with advanced designs (a prospective cohort study and randomized controlled trials) and adequate participant numbers would be helpful in identifying the determinants of successful surgical repair. Moreover, the authors would like to recommend the development of a standardized research tool for evaluating successful recovery assessment so that future researchers can use a consistent tool.

## Supporting information

**S1 Checklist. PRISMA 2020 checklist.**
(DOCX)

**S1 Text. Search strategy.**
(DOCX)

**S1 Table. Risk of bias assessment.**
(DOCX)

**S2 Table. Study characteristics.**
(DOCX)

**S1 Data. Data supporting the study.**
(XLSX)

## Acknowledgments

We would like to thank SuperScript Writing and Editing for the assistance.

## Author Contributions

**Conceptualization:** Liknaw Bewket Zeleke, Alec Welsh, Gedefaw Abeje, Marjan Khejahei.

**Data curation:** Liknaw Bewket Zeleke, Alec Welsh, Gedefaw Abeje, Marjan Khejahei.

**Formal analysis:** Liknaw Bewket Zeleke, Alec Welsh, Gedefaw Abeje, Marjan Khejahei.

**Methodology:** Liknaw Bewket Zeleke, Alec Welsh, Gedefaw Abeje, Marjan Khejahei.

**Software:** Liknaw Bewket Zeleke, Gedefaw Abeje, Marjan Khejahei.

**Supervision:** Alec Welsh, Gedefaw Abeje, Marjan Khejahei.

**Writing – original draft:** Liknaw Bewket Zeleke, Marjan Khejahei.

**Writing – review & editing:** Liknaw Bewket Zeleke, Alec Welsh, Gedefaw Abeje, Marjan Khejahei.

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
