## [Decision Letter · Decision Letter 0]

12 Apr 2024

PONE-D-24-01838Proportions and determinants of successful surgical repair of obstetric fistula in low- and middle-income countries: A systematic review and meta-analysisPLOS ONE

Dear Dr. Zeleke,

Thank you for submitting your manuscript to PLOS ONE. After careful consideration, we feel that it has merit but does not fully meet PLOS ONE’s publication criteria as it currently stands. Therefore, we invite you to submit a revised version of the manuscript that addresses the points raised during the review process.

 Please submit your revised manuscript by May 27 2024 11:59PM. If you will need more time than this to complete your revisions, please reply to this message or contact the journal office at plosone@plos.org. Please include the following items when submitting your revised manuscript:A rebuttal letter that responds to each point raised by the academic editor and reviewer(s). You should upload this letter as a separate file labeled 'Response to Reviewers'.A marked-up copy of your manuscript that highlights changes made to the original version. You should upload this as a separate file labeled 'Revised Manuscript with Track Changes'.An unmarked version of your revised paper without tracked changes. You should upload this as a separate file labeled 'Manuscript'.If applicable, we recommend that you deposit your laboratory protocols in protocols.io to enhance the reproducibility of your results. Protocols.io assigns your protocol its own identifier (DOI) so that it can be cited independently in the future. For instructions see: https://journals.plos.org/plosone/s/submission-guidelines#loc-laboratory-protocols. Additionally, PLOS ONE offers an option for publishing peer-reviewed Lab Protocol articles, which describe protocols hosted on protocols.io. Read more information on sharing protocols at https://plos.org/protocols?utm_medium=editorial-email&utm_source=authorletters&utm_campaign=protocols.

We look forward to receiving your revised manuscript.

Kind regards,

Ganesh Dangal, MD, FICS, FRCOG

Academic Editor

PLOS ONE

Journal Requirements:

2. In the online submission form, you indicated that data supporting this study are available from the corresponding author up on reasonable request.

Reviewers' comments:

Reviewer's Responses to Questions

**Comments to the Author**

1. Is the manuscript technically sound, and do the data support the conclusions?

Reviewer #1: Yes

Reviewer #2: Yes

2. Has the statistical analysis been performed appropriately and rigorously? 

Reviewer #1: Yes

Reviewer #2: Yes

3. Have the authors made all data underlying the findings in their manuscript fully available?

Reviewer #1: Yes

Reviewer #2: Yes

4. Is the manuscript presented in an intelligible fashion and written in standard English?

Reviewer #1: Yes

Reviewer #2: Yes

5. Review Comments to the Author

Reviewer #1: Well written and acceptable to publish.

Relevant review has been performed. Methodology is well explained. Study variables are specified as per clinical significance for the clinical outcome. Statistical tests applied are appropriate.

Reviewer #2: Thank you for offering this opportunity to review manuscript by Zeleke et al. titled "Proportions and determinants of successful surgical repair of obstetric fistula in low and middle-income countries: A systematic review and meta-analysis". It is interesting topic of the moment and I carefully reviewed the manuscript.

I would like to applaud authors for the amount of work they put together in this manuscript. With following minor revisions manuscript can be accepted for further processing

I have few minor comments

1. Authors are recommended to check the publisher guideline and edit the draft adhering to the guideline: in particular formatting, reference management (style, and uniformity) etc. https://journals.plos.org/plosone/s/submission-guidelines

2. I believe this statement may be partly untrue (Introduction section, last paragarph) “However, to the authors' knowledge, no systematic review and meta-analysis study has been conducted on the proportion of successful surgical repair outcomes and determinants of surgical repair outcomes in low- and middle-income countries.”

3. Discussion: It is recommended to have rigorous discussion taking consideration of the systematic reviews and meta-analysis on this topic. There are handful of meta-analysis authors can look for and compare and contrast the findings and have robust discussions. Such meta-analysis can be those including earlier publications or recent data (eg. https://pubmed.ncbi.nlm.nih.gov/34981462/) including data from publications after significant improvements in contemporary practice of fistula repairs (https://link.springer.com/article/10.1007/s11934-017-0708-5 ).

6. PLOS authors have the option to publish the peer review history of their article (what does this mean?). If published, this will include your full peer review and any attached files.

Reviewer #1: **Yes: **Gehanath Baral

Reviewer #2: No

---

## [Author Response · Author response to Decision Letter 0]

15 Apr 2024

I have attached a point-by-point response letter. Please find the letter for the details of the responses.

---

## [Editor Report · Decision Letter 1]

18 Apr 2024

Proportions and determinants of successful surgical repair of obstetric fistula in low- and middle-income countries: A systematic review and meta-analysis

PONE-D-24-01838R1

Dear Dr. Zeleke,

We’re pleased to inform you that your manuscript has been judged scientifically suitable for publication and will be formally accepted for publication once it meets all outstanding technical requirements.

Kind regards,

Ganesh Dangal, MD, FICS, FRCOG

Academic Editor

PLOS ONE
---

## [Editor Report · Acceptance letter]

26 Apr 2024

PONE-D-24-01838R1 

PLOS ONE

Dear Dr. Zeleke, 

I'm pleased to inform you that your manuscript has been deemed suitable for publication in PLOS ONE. Congratulations! Your manuscript is now being handed over to our production team.

Kind regards, 

on behalf of

Prof. Dr. Ganesh Dangal 

Academic Editor

PLOS ONE